# Eradication of Human Immunodeficiency Virus Type-1 (HIV-1)-Infected Cells

**DOI:** 10.3390/pharmaceutics11060255

**Published:** 2019-06-01

**Authors:** Nejat Düzgüneş, Krystyna Konopka

**Affiliations:** Department of Biomedical Sciences, Arthur A. Dugoni School of Dentistry, University of the Pacific, 155 Fifth Street, Room 412, San Francisco, CA 94103, USA; kkonopka@pacific.edu

**Keywords:** suicide gene therapy, CRISPR/Cas9, broadly neutralizing antibodies, cytotoxic liposomes, lentivirus

## Abstract

Predictions made soon after the introduction of human immunodeficiency virus type-1 (HIV-1) protease inhibitors about potentially eradicating the cellular reservoirs of HIV-1 in infected individuals were too optimistic. The ability of the HIV-1 genome to remain in the chromosomes of resting CD4+ T cells and macrophages without being expressed (HIV-1 latency) has prompted studies to activate the cells in the hopes that the immune system can recognize and clear these cells. The absence of natural clearance of latently infected cells has led to the recognition that additional interventions are necessary. Here, we review the potential of utilizing suicide gene therapy to kill infected cells, excising the chromosome-integrated HIV-1 DNA, and targeting cytotoxic liposomes to latency-reversed HIV-1-infected cells.

## 1. Antiretroviral Therapy

Following cell entry, human immunodeficiency virus type 1 (HIV-1) integrates its genome into the host cell chromosome. While some infected cells produce new virions, others are infected latently and do not express any viral envelope glycoproteins (Env; gp120/gp41) on their surface [1,2]. Treatment with antiviral drugs can inhibit the binding of the viral glycoproteins to the co-receptors on host cells (e.g., maraviroc), the fusion of the viral membrane with host cells (e.g., enfuvirtide), reverse transcription of the RNA genome of the virus (e.g., abacavir, lamivudine), integration of the reverse transcribed, double-stranded DNA into the host cell chromosome (e.g., raltegravir), and the viral protease involved in viral polyprotein cleavage and virus maturation (e.g., indinavir, darunavir). These treatments, however, do not eliminate the source of the virus, which is the HIV provirus integrated into the cellular chromosome.

Various therapeutic approaches have been tried in an attempt to eradicate the source of the virus. Soon after the introduction of protease inhibitors for HIV therapy, a mathematical analysis of the decay of blood levels of the virus following the co-administration of nelfinavir, zidovudine, and lamivudine suggested that cell-free virions and productively infected CD4+ T cells would be eliminated in less than two months [3]. The analysis also suggested that lymphocytes latently infected with an infectious provirus could be “completely eliminated after 2.3–3.1 years of treatment with a 100%-inhibitory antiretroviral regimen.” Nevertheless, the possibility was raised that longer treatment periods might be needed because of the “possible existence of undetected viral compartments or sanctuary sites”, as well as the persistence of infected mononuclear cells that could not be activated to produce virions [3]. Therapeutic interventions have also included early antiretroviral treatment during seroconversion, structured treatment interruptions, and targeted toxins; however, these approaches have not been able to eradicate the virus [4,5]. 

## 2. HIV-1 Latency

HIV-1 remains latent in some infected cells. These cells do not express any viral proteins on their surface, and they do not present any peptides in association with Class I major histocompatibility (MHC) molecules [6,7,8]. These latently infected cells, widely believed to be memory CD4+ T cells, and possibly cells of the monocyte-macrophage lineage [9,10,11], are hidden from recognition by the cellular immune system and render HIV infection intrinsically incurable with current antiretroviral therapy alone [12]. It is thought that the low-level viremia that continues despite therapy and the short-term viral blips (RNA below 200 copies/mL) do not depend on the presence of new drug-resistance mutations due to active replication but rather arise from viral release from stable reservoirs [12,13]. 

HIV latency is a multifactorial process. In most cases, the HIV-1 proviral DNA, reverse transcribed from the viral RNA genome, integrates into the host cell genome in regions that are being transcribed actively. It is ironic that most latent proviral DNA in patients who are on combination antiretroviral therapy (cART) and have suppressed viral replication are found in actively transcribed segments of the cellular chromosomes [13]. Several mechanisms of transcriptional interference may be involved in suppressing the expression of the proviral DNA: (i) the cytoplasmic sequestration of transcription factors, including NF-κB and NFAT, may inhibit viral gene expression; (ii) if the host promoter is located upstream of the provirus, the RNA polymerase (Pol II) may “read through” the HIV-1 promoter (5′-LTR) and displace the transcription factors necessary for viral transcription; (iii) if the proviral DNA has integrated into the host chromosome in the opposite orientation of the host gene, the RNA Pol II complexes of the provirus and the host may collide, and RNA transcription stops; (iv) resting CD4+ T cells have low levels of Cyclin T1, which would otherwise form P-TEFb and also play a role in HIV-1 transcription and Tat-mediated transactivation—this would therefore result in a lack of transcription of proviral DNA; (v) methylation of DNA and compaction of chromatin may also contribute to transcriptional silencing and hence HIV-1 latency [13]. 

A groundbreaking study by Lehrman et al. [14] showed that inhibition of histone deacetylase (the chromatin remodeling enzyme that helps maintain the latency of integrated HIV-1) by valproic acid, together with cART supplemented with enfuvirtide (ENF), reduced the frequency of resting cell infection. This finding suggested that, with new approaches, it may be possible to reduce or eliminate the reservoir of HIV in infected individuals. A subsequent study employing raltegravir or enfuvirtide in addition to standard cART and valproic acid in a slightly larger patient pool showed, however, that there was no effect on the low-level viremia measured by single-copy plasma HIV RNA [15]. This approach relied on cell death upon activation of the latently infected cells, or by recognition and destruction by the immune system. Margolis et al. [16] indicated that “a major approach to HIV eradication envisions antiretroviral suppression, paired with targeted therapies to enforce the expression of viral antigen from quiescent HIV-1 genomes, *and immunotherapies to clear latent infection*.” Sengupta and Siliciano [17] reiterated this point, stating “neither viral cytopathic effects nor CTL-mediated lysis may occur upon latency reversal *without additional interventions*.” In this mini-review, we describe most of these “additional interventions”, including some approaches from our laboratory.

## 3. Latency Reversal

Richman et al. [9] emphasized that latency is likely to be established and maintained by blocks at different steps in the HIV-1 replicative cycle, which may complicate efforts to eradicate these cells. One of these blocks was inhibited by the administration of the histone deacetylase inhibitor, vorinostat, to HIV-infected patients, as shown by cellular acetylation and an increase in HIV RNA expression in resting CD4+ cells [18]. Thus, the latency of HIV-1 in resting CD4+ cells in patients can be reversed by a tolerable dose of vorinostat. The low number of latently infected cells in patients has hindered studies on HIV-1 activation. Latently infected T cell lines are thus useful in experiments identifying the agents that can activate latent HIV-1. Another problem with the activation of latently infected cells is the percentage of cells that are actually induced to synthesize viral proteins. The site of chromosomal insertion of HIV-1 DNA affects the response to activating agents; for example, phytohemagglutinin and vorinostat reactivated proviruses at distinct genomic locations [19]. Since the activation of the cells will produce infectious virions, it will be necessary to concomitantly employ a cART regimen used in the studies of Lehrman et al. [20] (2005) and Archin et al. [15].

Latency reversing agents include various protein kinase C activators, such as diacylglycerol lactones [21], anti-tumor-promoting phorbol esters [4,22] histone deacetylase inhibitors [23], interleukin-7 [14], toll-like receptor-1 and-7 agonists [24], bryostatin analogs [25] and other compounds [26,27]. Spina et al. [26] compared the response to numerous activators of primary T cell models, J-Lat cell lines, and patient-derived infected cells, and found that the different cellular systems responded differently to the activators than the latently infected T cells obtained from patients. Nevertheless, protein kinase C agonists and phytohemagglutinin were able to activate latent HIV in all the cellular models developed in different laboratories. A library of marine natural products was used to identify four latency reversing agents, including aplysiatoxin, which induced the expression of HIV-1 provirus in cell lines and primary cell models at concentrations 900-fold lower than that of prostratin without significant effects on cell viability [28]. The expression of provirus-derived RNA in single cells upon the reversal of latency—even before the synthesis of viral proteins—may be useful in delineating the mechanisms of these agents [29].

Another approach to reversing latency is the use of the CRISPR/Cas9-derived systems [SunTag and synergistic activation mediator (SAM) systems] that recruit several transcriptional activation domains to an optimal target region within the HIV 5′ long terminal repeat (LTR) [30]. Transcriptional activation of proviral genomes was observed in various latently infected cell lines at levels comparable to or higher than treatment with established latency reversal agents and led to the production of infectious virions. A similar system that comprised an RNA-guided dCas9-VP64 activator targeting the junction between two NF-κB transcription factor-binding sites within the LTR enhancer region induced transcriptional activation of latent HIV-1 infection in all latency models tested [31].

In the “shock and kill” studies described above, it was expected that HIV-1-specific CD8+ cytotoxic T lymphocytes would recognize HIV-1 antigens on the activated CD4+ T cells and kill them. However, some latency reversal agents were found to also adversely affect the function of CD8+ cells [32,33]. Another consideration in this approach, which is also relevant to the other methods described below, is the potential of stimulating a deleterious, systemic inflammatory response [33]. Huang et al. [34] showed that following the ex vivo exposure of CD4+ T lymphocytes from ART-treated individuals to several latency-reversing agents and autologous CD8+ T lymphocytes reduced cellular HIV DNA but could not deplete the replication-competent virus. This observation may indicate that cells containing replication-competent HIV are resistant to cytotoxic T cells.

## 4. Suicide Gene Therapy

The insertion of potentially cytotoxic genes that can be activated by characteristic proteins of HIV-1 may be a useful approach to eliminating HIV-1-infected cells. Plasmids expressing the diphtheria toxin A-fragment (DT-A) under the control of HIV-1 LTR sequences (−167 to +80) could be *trans*-activated by the viral Tat protein, but the toxin gene was also expressed to some extent without Tat [35]. When *cis*-acting negative regulatory elements from the *env* region of the viral genome were incorporated in the 3’ untranslated region of these plasmids, basal expression from the LTR was minimized. The DT-A gene could then be *trans*-activated at a maximal level in the presence of both Tat and Rev proteins from the virus. CD4+ H9 cells that were transduced with a retrovirus to express this gene construct and then infected with HIV-1 could limit HIV-1 production [36]. The intracellularly “immunized” cells were protected against clinical HIV-1 strains up to 59 days [37]. Co-transfection of HeLa cells with the plasmid expressing DT-A and the proviral HIV-1 clone, HXBΔBgl, completely inhibited virus production by the cells [38].

Our laboratory is working on developing an HIV-1-specific promoter that minimizes basal expression (e.g., in the absence of the HIV-1 transactivator, Tat) and that drives the expression of suicide genes that would induce cell death specifically in HIV-1-infected cells. As an initial attempt at modifying the LTR, we generated several clones with deletions in parts of the LTR (Figure 1). For example, LTR2 had the modulatory region deleted, whereas in LTR3, the NF-κB binding site was deleted. In a model system employing Tat-expressing HeLa cells and luciferase-expressing, wild-type, and mutant LTR-driven plasmids, we showed that the mutant LTR2, from which the modulatory elements at the 5′ end of the LTR were deleted, was 100-fold more responsive to the presence of Tat (which would only be present in infected cells that are actively producing the virus) than in control cells not expressing any Tat (e.g., uninfected cells or quiescent, latently-infected cells) [39] (Figure 2). The LTR2 promoter may now be used to drive the expression of the herpes simplex virus thymidine kinase (HSV-*tk*) gene preferentially in infected cells, resulting in cytotoxicity upon administration of the anti-herpesvirus drug, ganciclovir (Figure 3). To enable this construct to be delivered to all HIV-1-infected cells, it would most likely have to be incorporated into an HIV-1-based lentiviral vector. The advantage of this system over the DT-A-based plasmid described above is that it can be turned on and off. Thus, “treatment” can be initiated by the administration of ganciclovir and interrupted by ceasing it. We previously studied cytotoxicity induced by the HSV-*tk* system in oral squamous cell carcinoma and cervical carcinoma cells [40,41,42,43], as well as in an animal model of oral cancer [44,45].

There is still a finite amount of luciferase expression from LTR1, LTR2, and LTR3 in the absence of Tat (Figure 2). Further genetic engineering of the HIV-1 promoter will be necessary to eliminate even this basal level of gene expression.

Garg and Joshi [46] cloned the *tk* gene into the vector, pNL-GFPRRESA, which includes the full LTR promoter that also expresses GFP in the presence of Tat. Following the selection of cells expressing GFP and treatment with ganciclovir, virus production and the number of virus-infected cells was reduced, demonstrating the feasibility of this system.

## 5. Excision of Chromosome-Integrated HIV-1 DNA

The modification and the development of the clustered regularly interspaced palindromic repeat (CRISPR)/Cas9 endonuclease system, originally identified in certain bacteria as an adaptive immune system, has led to a molecular tool that can target specific sequences in DNA [47,48]. This method could be particularly useful in cells that are not activated to produce HIV proteins and hence are not amenable to become targets of the immune system (as in the proposed “shock and kill” method), or in suicide gene activation by the Tat protein (*vide supra*). This system has been applied to alter the HIV-1 genome and block its expression. CRISPR/Cas9 components targeting various HIV1-derived sequences were transfected into T cells with integrated, LTR-driven GFP and TAR sequences. Upon stimulation of the cells, LTR-driven gene expression was inhibited significantly [49]. Sequence analysis confirmed that the targeted LTR and TAR sequences were cleaved.

Following the cleavage by the enzyme Cas9 of specific DNA sequences determined by intracellularly delivered guide RNA, the non-homologous end joining (NHEJ) machinery of the cells causes insertions and deletions (termed “indels”) at the cleaved site, thereby causing the impairment of DNA function at the site. In the case of HIV-1-infected cells, these indels can inactivate the virus, but they can also produce replication-competent virions that have a slightly different proviral DNA sequence. These sequences may now be resistant to recognition by the same guide RNA [50], demonstrating that the CRISPR/Cas9 system can both inactivate HIV-1 and generate mutant virus [51].

Transcription activator-like effector nucleases (TALENs) [52] were employed to target the LTR site used with the CRISPR/Cas9 system above. The intracellular introduction of mRNA encoding the specific TALEN caused about 80% of the target DNA to be removed [53].

The Cas9/guide RNA (gRNA) system was utilized to target the HIV-1 LTR U3 region and excised a 9709 bp fragment of integrated proviral DNA from its 5’ to 3’ LTRs [54]. This resulted in inactivation of viral gene expression and replication in various cell types, including a microglial cell line and a promonocytic cell line, without causing genotoxicity or off-target effects in the host cells. The same approach was applied to latently infected human CD4+ T-cells to cleave the chromosome-integrated proviral DNA, and whole-genome sequencing of the treated cells indicated that there was no effect on cell viability, cell cycle, and apoptosis. The co-expression of Cas9 and the targeting gRNAs in cells from which HIV-1 had been eradicated protected the cells against *de novo* HIV-1 infection [55]. In a study with transgenic rodents with the HIV-1 genome, a short version of the Cas9 endonuclease was used in conjunction with a multitude of gRNAs that targeted the viral 5’-LTR and the *gag* gene and delivered in an adeno-associated virus [56]. This treatment resulted in the generation of a 978 bp HIV-1 DNA fragment in various organs and in circulating lymphocytes, indicating the cleavage of part of the proviral DNA.

## 6. Cytotoxic Liposomes Targeted to HIV-1-Infected Cells

In addition to the delivery of an HIV-1-activated suicide gene into latently-infected cells, our laboratory is focusing on the killing of such cells by targeting liposomes encapsulating cytotoxic drugs to infected cells whose latency has been reversed and that now express Env on their surface. For this purpose, we are coupling broadly neutralizing anti-Env antibodies (bNAb) [57,58,59], CD4-immunoadhesin [60], or CD4-derived peptides [61] as ligands to target the activated cells (Figure 4). Such liposomes are expected to be internalized, as shown for liposomes targeted to cancer cells [62,63] (*vide infra*) and kill the infected cells. To prevent an immune reaction to the antibodies, they can be engineered to be “humanized” for eventual clinical use, as in the case of a number of antibody-based drugs, including anti-HER2 [64].

Sterically stabilized liposomes containing poly(ethylene glycol) (PEG)-conjugated lipids and loaded with the cytotoxic DNA-intercalating anticancer drug, doxorubicin, are currently approved for the treatment of Kaposi’s sarcoma, ovarian cancer, breast cancer, and multiple myeloma [65]. These liposomes have prolonged circulation in the bloodstream and can extravasate into tissues, including lymph nodes [66]. Liposomes administered subcutaneously are cleared via the local lymph nodes [66,67], thus localizing in tissues where HIV-1 is either replicating or hiding in latently infected cells [68,69,70]. Subcutaneous injection of liposomes carrying indinavir resulted in a 21–126 fold higher accumulation of the drug in all tissues compared to the free drug [71]. Indinavir delivered subcutaneously in liposomes to HIV-1-infected macaques localized in lymph nodes and caused a significant reduction in viral load [72]. Liposomes encapsulating the HIV-1 protease inhibitor, L-689,502, reduced the EC_50_ of the drug by 3-5-fold in infected macrophages [73]. The protease inhibitor PI1 encapsulated in liposomes targeted to gp120 expressed on infected cells via the antibody F105 had a 10-fold higher anti-HIV activity than the free drug at 100 nM [74]. However, it should be noted that, although the activity of antiviral agents can be enhanced by delivery in liposomes, and passive targeting to lymph nodes may reduce the need for daily administration of the drugs, this approach will not lead to the eradication of HIV-1-infected cells.

Nevertheless, liposomes containing cytotoxic drugs and targeted via anti-HER2 (ErbB2) monoclonal antibody fragments have been utilized in cancer chemotherapy. For example, they enhanced doxorubicin uptake in HER2-overexpressing cells in culture by up to 700-fold and resulted in tumor regression in five different tumor xenograft animal models [62], indicating their superior ability to kill tumor cells. Doxorubicin liposomes targeted to cell surface CD44 receptors on B16F10 melanoma cells had a 5-6-fold higher rate constant of cell killing than the free drug for a given amount of intracellular doxorubicin [63], showing the effectiveness of targeting. Sterically stabilized liposomes can also be rendered pH-sensitive to facilitate or enhance the intracellular delivery of cytotoxic drugs [75,76].

Since latently infected cells do not express the viral glycoproteins on their surface, they need to be activated by “latency reversing agents” to produce HIV-1 and express Env and hence will be recognizable by the targeted liposomes. These liposomes do not have to be administered for prolonged periods of time, since they will eliminate HIV reservoirs, unlike current treatment modalities with lifelong administration of antiviral agents. Thus, the inconvenience of subcutaneous, intravenous, intraperitoneal, or spinal delivery of liposomes is likely to be tolerable by patients who are likely to be cured of their HIV infection, perhaps after a series of injections.

The activated, previously latently infected cells are thus expected to express the HIV-1 Env protein on their surface, be recognized by the targeted cytotoxic liposomes, and be killed as a result of the intracellular delivery of the cytotoxic drug.

The fact that doxorubicin encapsulated in sterically stabilized liposomes is already approved for clinical use in the treatment of various cancers supports the feasibility of our approach [65,77]. An additional route of liposome administration for lymph node accumulation is intraperitoneal injection [78]. Anti-HLA-DR-bearing sterically stabilized liposomes accumulate in the lymph node cortex following subcutaneous injection [79]. Liposomes can also be injected into the spinal cord, as demonstrated in an animal model [80], enabling them to reach HIV-1-infected macrophages/microglia in the central nervous system [81]. Although this appears to be a difficult procedure, it is considerably more applicable than complete eradication of the immune system for bone marrow transplantation of HIV-1-resistant CCR5Δ32/Δ32 hematopoietic stem cells, a method that was applied in curing the “Berlin patient” and the “London patient” [82,83]. Therefore, after we demonstrate that targeted cytotoxic liposomes can specifically eliminate HIV-1-infected cells in culture, it will be possible to apply this method in vivo in HIV-1 infection models and eventually in patients.

## 7. Concluding Remarks

HIV-1 latency, the ability of the HIV-1 genome to remain in the chromosomes of resting CD4+ T cells and macrophages without being expressed, has been an important challenge in attaining HIV-1 remission and an eventual cure. It is astounding that a virus with just a few genes has evolved a way to reverse transcribe and integrate its genome into host cell chromosomes, but that virologists and molecular biologists throughout the world have not yet come up with an effective solution to excise or inactivate the viral genome or to specifically kill the infected cells. One possible reason for this is that the prevailing dogma in the scientific community soon after the identification of HIV-1 and HIV-2 as the etiologic agents of acquired immune deficiency syndrome (AIDS) was that it was impossible to cure HIV/AIDS and that our efforts should be focused on preventing viral replication. Nevertheless, in 1993, we proposed to use oligonucleotide-conjugated endonucleases [84,85] to cleave the chromosome-integrated HIV-1 provirus. In 2004, we also proposed to use triple-helix forming oligonucleotides [86,87] with an intervening spacer that would hybridize with the viral LTRs at the beginning and the end of the provirus, thereby connecting the two LTRs via the spacer and potentially inducing DNA repair mechanisms that would remove the looped DNA. We expect that the approaches we reviewed here (suicide gene therapy to kill infected cells, excision of chromosome-integrated HIV-1 DNA, and cytotoxic liposomes targeted to latency-reversed HIV-1-infected cells) will be developed further to be able to treat infected patients.

## Figures and Tables

**Figure 1 pharmaceutics-11-00255-f001:**
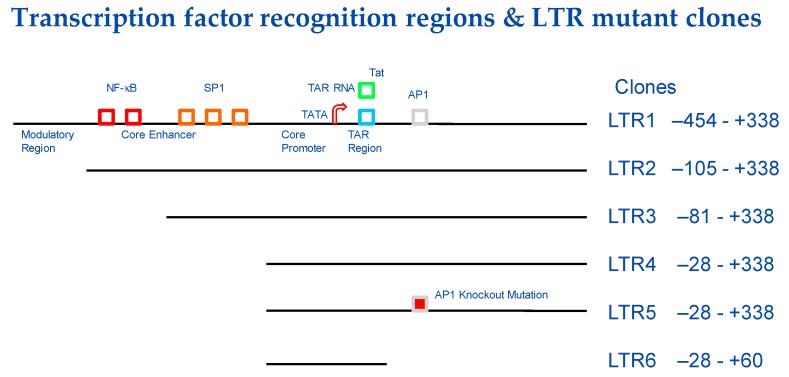
Transcription factor recognition regions of human immunodeficiency virus type-1 (HIV-1) long terminal repeat (LTR) and mutated LTR sequences used in HIV-1-specific gene expression studies shown in Figure 2.

**Figure 2 pharmaceutics-11-00255-f002:**
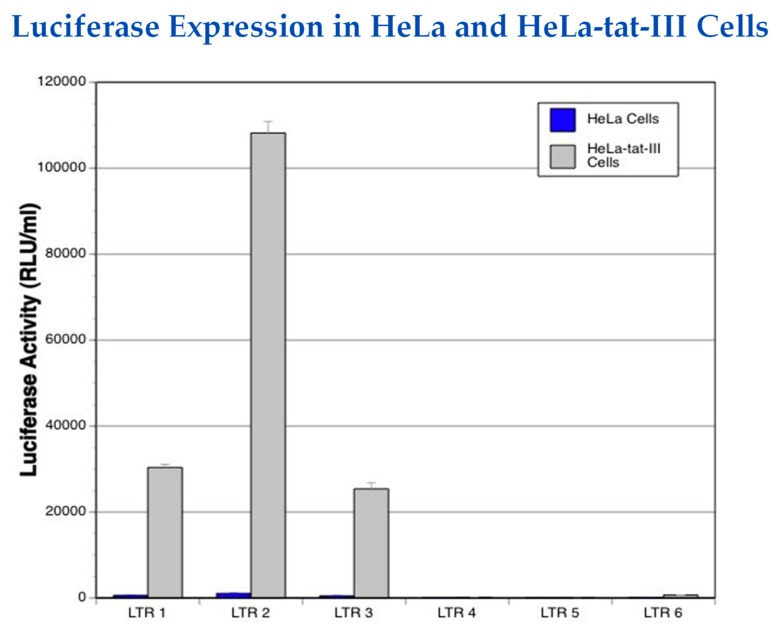
Comparison of luciferase gene expression from wild type LTR and LTR mutant clones (shown in Figure 1) in HeLa cells and HeLa-tat-III cells that constitutively express the HIV-1 Tat protein (data from [39]).

**Figure 3 pharmaceutics-11-00255-f003:**
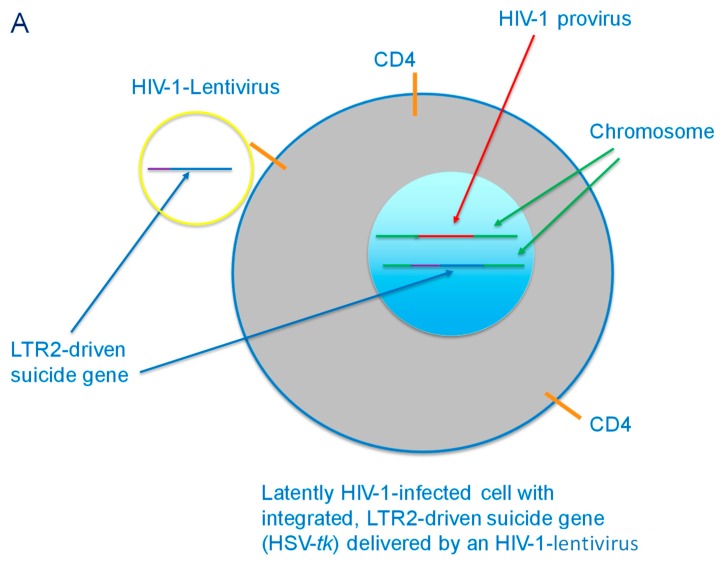
(**A**) HIV-1-lentivirus delivery of LTR2-driven suicide gene (HSV-*tk*) to a latently HIV-1-infected CD4+ cell. The suicide gene is then integrated into the chromosome of the cell. (**B**) Latency reversing agent-activated HIV-1-infected cell with integrated LTR2-driven suicide gene. The cell produces viral proteins, including Tat, which activates the LTR2 to express HSV-*tk*, which then monophosphorylates ganciclovir that has been delivered to the cell.

**Figure 4 pharmaceutics-11-00255-f004:**
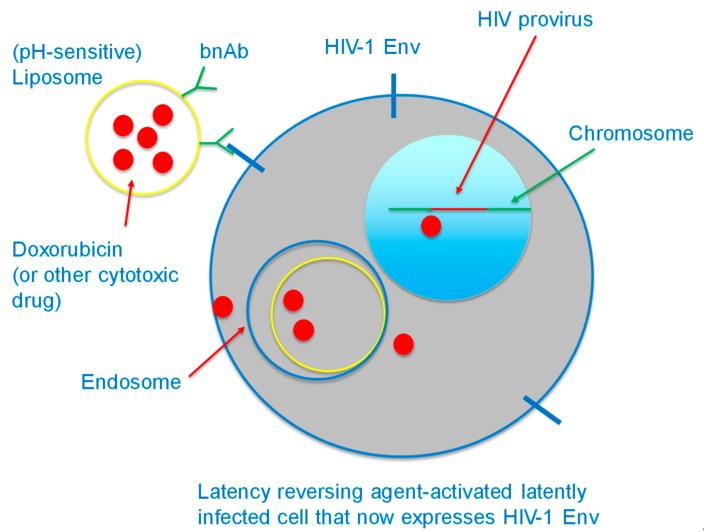
Cytotoxic liposome targeted to cell surface Env, which is expressed following treatment of a latently infected cell with a latency reversing agent. The targeting ligand is a broadly neutralizing anti-Env antibody. The liposome is endocytosed after binding to cell surface Env. The liposome may be engineered to be pH-sensitive so as to destabilize the endosome membrane at mildly acidic pH achieved in the endosome lumen and to enhance drug delivery to the cytoplasm.

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
