# Peer review of "Eradication of Human Immunodeficiency Virus Type-1 (HIV-1)-Infected Cells"

_pharmaceutics, 2019, doi:10.3390/pharmaceutics11060255_

Round 1
Reviewer 1 Report
In this manuscript, Drs. Düzgüneş and Konopka described several methods aimed at eradication of latently HIV-infected cells. The topics discussed here will be interesting to readers. However, the writing of the manuscript is somewhat undetailed and weak in terms of logical structure and story. The authors’ contributions to this field should be emphasized. Below is a list of comments to be addressed.
1. In page 1, in the 2nd paragraph of Section 1, the authors state that the mathematical modeling by Perelson et al. (1997) estimated the duration of antiretroviral therapy necessary to eliminate the integrated HIV proviral copies in HIV-infected compartments (productively infected cells, long-lived infected cells, and latently infected lymphocytes) as 2.3-3.1 years. Therefore, the authors state that Perelson et al. failed to predict the persistence of viral reservoir for longer periods. Here, I suppose it is important to explain the reason for the failure of their prediction. It seems to me that Perelson et al. failed in their prediction because they underestimated the half-lives of HIV-infected cells including latently infected lymphocytes in the mathematical model. However, please check the last sentence of the abstract of Perelson et al. (1997) that carefully states, “To eradicate HIV-1 completely, even longer treatment may be needed because of the possible existence of undetectable compartments or sanctuary sites”. This means that in the discussion of the paper, Perelson et al. precisely predicted that there could be latently infected cell compartments with half-lives longer than reported at that time. The last part of the same article is also worth reading. So please revise the paragraph to convey the message of Perelson et al. (1997) correctly. In addition, it might be meaningful to provide information of the current estimate, if any, of how long a latently infected cell can survive in ART-treated patients.
2. In page 2 line 57, I see the word “NF- B”. I guess this is “NF-kappa B” but the Greek letter kappa is not visible here.
3. In page 3 line 108, I see the word “HXB Bgl”. Here it seems the Greek letter Delta is not visible.
4. In page 3 Section 4, the authors’ contributions to the field could be more emphasized.
a) In Gebremedhin al. 2012, the authors describe that the efficacy of LTR2 to respond to Tat is 100-fold higher than that of LTR. It would be helpful if the authors could explain the difference in the efficacies of LTR and LTR2 to kill latently HIV-infected cells.
b) The method using the Tat-sensitive LTR (LTR2) to express HSV/tk targeted by ganciclovir is interesting. I hope the authors could provide a figure with a schema to explain this in a more intuitive way.
c) Plus, I think the suicide gene therapy has been tested for long in cancers. More description with citations would be helpful to share the background of the authors’ studies.
d) LTR2 may be able to detect lower levels of Tat expression than LTR. However, I wonder whether LTR2 could be effective in latently HIV-infected cells with no expression of Tat. If yes, please emphasize it. If no or not yet tested, please help readers imagine how to test it on latently HIV-infected cells.
e) Garg & Joshi (2016) describes gene therapy based on the Tat-dependent HSV/tk + ganciclovir method. Did they use LTR2? If not, is it possible to use LTR2 in a lentivirus vector? This seems interesting.
5. In pages 3-4 Section 5, I think the cons and pros of CRISPR-Cas9 could be described in comparison to other methods. For example, CRISPR-Cas9 only requires the genome DNA and works regardless of whether HIV proteins are produced or not. This is contrary to many other methods (e.g. shock and kill, HSV/tk) that are dependent on the expression of the HIV gene. On the other hand, CRISPR-Cas9 may cause mutations at the site of cleavage, and this could result in the emergence of replication competent HIV variants resistant to the introduced guide RNA (Wang Z et al, 2016). I suppose these contrasting differences of CRSPR-Cas9 to other methods may be interesting to readers.
6. In pages 4-5 Section 6, the text could be improved.
a) In page 4 line 160, indinavir may be indicated as a protease inhibitor. Is it right?
b) In page 4, the 1st paragraph of Section 6 seems to be a mixture of evidences for cancer treatment and anti-HIV drug delivery. I suppose the evidences could be arranged in a more readable way to attract readers to the story of the manuscript.
c) In page 4 in the 2nd paragraph of Section 6, again, I think the description here on the authors’ contributions could be more stressed, possibly in some ways including a schema.
d) In page 4 in the 3rd paragraph of Section 6, the authors write, “These liposomes do not have to be administered for prolonged periods of time, since they will eliminate HIV reservoirs, unlike current treatment modalities with lifelong administration of antiviral agents.” If this has been demonstrated, please give a reference. If not, please better explain why liposomes are believed to eliminate HIV-infected cells. In this paragraph, I think the authors have proposed “shock and kill” methods and the “kill” methods are based on liposomes in Section 6 (and the kill methods described in Section 4 is LTR2-HSV/tk, is it right?).
e) In the same paragraph, the authors describe a number of latency-reversing agents. However, the writing of this paragraph or this section might ambiguate the meaning of the independent section for latency reversal (i.e. Section 3). I guess authors might wish to emphasize the importance of developing better “kill” methods to achieve viral eradication in vivo. If so, I could propose to better integrate descriptions associated with shock and kill methods and emphasize the authors’ ideas.
f) In page 5 lines 203-204, the authors write, “the demonstration that targeted cytotoxic liposomes can specifically eliminate HIV-1-infected cells in culture”. If this has been demonstrated, please give a reference. If this is yet to be demonstrated, please describe so.
7. In page 5 Section 7, the authors write, “It is astounding that a virus with just a few genes has “figured out” a way to reverse transcribe and integrate its genome into host cell chromosomes, but that virologists and molecular biologists throughout the world have not yet come up with a foolproof solution to excise or inactivate the viral genome, or to specifically kill the infected cells.” This may be right, but the sentence could be more scientific if the authors could further provide the reason for the delay to find a solution to eliminate infected cells. Plus, to further support the authors’ description, the HIV genome size (10 kbs) could be compared to other viruses (HSV: 150 kbps, VZV: 125 kpbs, EBV: 170 kbps, CMV: 236 kbps, HPV: 8 kbps) causing latent infection. Herpesviruses clearly have much larger genomes and cause episomal latency, while HIV can only cause proviral latency. It seems the mechanism for the HPV latency is currently less clear.
Author Response
Response to Reviewers
We greatly appreciate the time the Reviewers have taken and the very useful comments they have made. We are especially grateful for the positive and collegial tone of the reviews. We have revised the manuscript to include their suggestions and to make the recommended changes. The changes in the manuscript, including the References are marked in red type.
Reviewer 1
We have changed the introduction to include the comment by Perelson et al regarding the undetectable sanctuary sites for latent HIV-1.
2. The kappa is present in our manuscript, but appears to have been deleted during the generation of the pdf file by the journal.
3. The delta appears to have been deleted during the generation of the pdf file by the journal.
4. a. We have included a phrase indicating that modulatory regions of the LTR were removed to generate LTR2.
b. We have now included several figures showing the data on luciferase expression, the schema of the LTR mutants, and the activation of a suicide gene in cells that have been transduced by a recombinant lentivirus incorporating our constructs.
c. We have added references to some of our work on suicide gene therapy of oral and cervical cancer.
d. We have mentioned that even with LTR2 there is a basal expression of luciferase in the absence of Tat. This level of expression may be able to kill latently infected cells with no Tat expression; however, this may cause cytotoxicity to uninfected cells as well.
e. It appears that Garg and Joshi used only LTR as judged from their Figure 2. We have mentioned in the original manuscript the necessity to use an HIV-1-based lentiviral vector to be able to geliver the suicide gene construct into all HIV-infected cells.
5. We have added references to Zhang et al. (2016) and Wang et al. (2018) to discuss the possibility of generation of mutant virions.
6. a. We have added indinavir as an example of a protease inhibitor.
b. We have translocated the paragraph on the use of liposomes in cancer therapy to the end of the section on Cytotoxic Liposomes.
c. We have added a figure to depict the use of targeted cytotoxic liposomes to kill HIV-1 infected cells.
d. The use of targeted cytotoxic liposomes to eradicate HIV-1-infected cells has not been demonstrated. We have mentioned in the first paragraph of this section that “Such liposomes are expected to be internalized, as shown for liposomes targeted to cancer cells (Park et al., 2001; Eliaz et al., 2004) (vide infra), and kill the infected cells.“
e. We have moved the paragraph on different types of latency reversal agents to the section on Latency Reversal.
f. We have slightly reworded this section to clarify that once we demonstrate cell killing in culture, we will apply the technique in vivo.
7. Our statement in Concluding Remarks was to indicate partially that HIV-1 had evolved “ingenious” methods to integrate its genome into host chromosomes; but that this simple process could not be reversed by the ingenuity of virologists and molecular biologists. We have now provided two potential explanations for why it has taken so long to excise or inactivate the integrated HIV-1 provirus.
Reviewer 2 Report
This paper reviews several approaches for the eradication of HIV infected cells as an approach to virus cure. The authors discuss the limitation s of antiretroviral therapy and challenges posed by latent reservoirs of intact viral genome which can serve to initiate new rounds of infection once therapy is discontinued. These reservoirs are not accessible to antiviral attack and require activation by latency reversing drugs to activate viral expression of virus, making infected cells targets to cell killing (“Shock and kill”). The authors then proceed to discuss approaches in their lab which may provide further refinements to the shock and kill approach to HIV cure.
Although there is a reasonable review of possible approaches to virus eradication, the paper is seriously lacking in discussion of the limitations and concerns for these approaches. The Latency Reversing Agent (LRA) mentioned, such as the histone deacetylase inhibitors (HDACi), have not been demonstrated to induce sufficient viral antigen production for recognition of latently infected cells by HIV specific CD8+ T-Cells in vivo, and have failed to achieve effective in vivo exposure of virus to cell killing clinical trials to date. (Clutton and Jones Frontiers in Immunology 9: June 2108.). Furthermore, latent cells infected with replication competent virus appear to be resistant to killing even in the face of powerful LRAs and potent HIV specific CD8+TR cells (Huang et al, 2018, J Clin Invest 128: 876-89.) Testing of LRAs in viral suppression assays have demonstrated detrimental effects on T-cell toxicity or inhibition of T cell proliferation, which will limit their utility. Another major hurdle is the difficulty or reactivating all cells harboring HIV replication competent cells, as only a small subset of cells can be activated, leaving the bulk of the latent reservoir intact. A review of current literature on this topic is lacking.
The use of cytotoxic liposomes for delivery of anticancer drug to specific target sites may be a promising new approach in cancer therapy. The authors propose to target liposomes encapsulating cytotoxic rugs to activity HIVI producing cells expressing env on their surface using broadly neutralizing antibodies. However the feasibility of such a n approach has not been demonstrated, or whether the antibody will effectively deliver the drug to the target tissue.
The proposed studies do not address the limitations and real obstacles that would needed for virus eradication, and need to include more current literature on these topics.
Author Response
Response to Reviewers
We greatly appreciate the time the Reviewers have taken and the very useful comments they have made. We are especially grateful for the positive and collegial tone of the reviews. We have revised the manuscript to include their suggestions and to make the recommended changes. The changes in the manuscript, including the References are marked in red type.
Reviewer 2
1. “the paper is seriously lacking in discussion of the limitations and concerns for these approaches.”
a. Regarding latency reversal, even the original manuscript included the statement “neither viral cytopathic effects nor CTL-mediated lysis may occur upon latency reversal without additional interventions.”
b. The revised manuscript includes the statement “Another problem with the activation of latently infected cells is the percentage of cells that are actually induced to synthesize viral proteins. The site of chromosomal insertion of HIV-1 DNA affects the response to activating agents…”
c. Regarding suicide gene therapy, we state “There is still a finite amount of luciferase expression from LTR1, LTR2 and LTR3 in the absence of Tat (Figure 2). Further genetic engineering of the HIV-1 promoter will be necessary to eliminate even this basal level of gene expression.
d. Regarding excision, we have included the sentences “. In the case of HIV-1-infected cells, these indels can inactivate the virus, but they can also produce replication-competent virions that now have a slightly different proviral DNA sequence. These sequences may now be resistant to recognition by the same guide RNA (Zang et al., 2016) , demonstrating that the CRISPR/Cas9 system can both inactivate HIV-1 and generate mutant virus (Wang et al., 2018).
e. Regarding cytotoxic lipposomes, we mention the following: “Although this appears to be a difficult procedure, it is considerably more applicable than complete eradication of the immune system for bone marrow transplantation of HIV-1-resistant, CCR5D32/D32 hematopoietic stem cells, a method that was applied in curing the “Berlin patient” and the “London patient” (Hütter et al., 2009; Gupta et al., 2019).”
2. “Latency Reversing Agent (LRA) mentioned, such as the histone deacetylase inhibitors (HDACi), have not been demonstrated to induce sufficient viral antigen production for recognition of latently infected cells by HIV specific CD8+ T-Cells in vivo, and have failed to achieve effective in vivo exposure of virus to cell killing clinical trials to date.”
We have discussed the references mentioned by the Reviewer (Clutton and Jones; Huang et al).
3. “Another major hurdle is the difficulty or reactivating all cells harboring HIV replication competent cells, as only a small subset of cells can be activated, leaving the bulk of the latent reservoir intact. A review of current literature on this topic is lacking.”
We have discussed two additiona systems for reactivation demonstrated by Bialek et al. (2016) and Saayman et al. (2016).
4. “However the feasibility of such an approach has not been demonstrated, or whether the antibody will effectively deliver the drug to the target tissue.”
One of the purposes of this review was to introduce two new approaches to eradicating HIV-1-infected cells. Indeed, the use of cytotoxic, targeted liposomes has not been demonstrated, but will be evaluated once our grant application in this area is funded.
5. “The proposed studies do not address the limitations and real obstacles that would needed for virus eradication, and need to include more current literature on these topics.”
We disagree with Reviewer 2 on this point. We have cited current literature on this topic and have cited the conclusions of leaders in the field; for example, “Margolis et al. (2016) indicated that “a major approach to HIV eradication envisions antiretroviral suppression, paired with targeted therapies to enforce the expression of viral antigen from quiescent HIV-1 genomes, and immunotherapies to clear latent infection.” Sengupta & Siliciano (2018) reiterated this point, stating “neither viral cytopathic effects nor CTL-mediated lysis may occur upon latency reversal without additional interventions.”
Reviewer 3
“However, my main concern is the lack of a balanced perspective presenting the critical aspects that hamper HIV-1 cure nowadays, the challenges ahead to make these strategies work in the clinic, and most importantly, the limited effect of viral reactivation agents in vivo that precludes effective HIV-1 reactivation.”
In the revised manuscript, we have indicated some of the problems with latency reversal approaches, such as the adverse effects on CD8+ cells and non-specific inflammatory responses.
2. “effective immune interventions, have to be clearly stated and further discussed in each section.”
Our review is not focused on immune interventions, but provides alternatives to the expectation that CD8+ T cells would kill activated, infected cells. Of course, the generation of an effective vaccine (which has unfortunately been in the works since the mid-1980s, without success) may help eradicate some infected cells and prevent the infection of new cells.
3. “All these sentences and the animal models in which they have been tested should be clearly indicated.”
We have clarified the fact that the studies we have cited were done in animal models.
4.1. “…, it provides the first conceptual proof of principle that eliminating HIV infected cells can lead to an actual HIV-1 cure/remission.”
We have briefly discussed the Berlin and London patients and provided references to these cases at the end of the Cytotoxic Liposomes section.
4.2. “An illustration depicting the action of LRAs, cytotoxic liposomes, suicide gene therapy and excision of Chromosome-integrated HIV-1 DNA could help to summarize approaches discussed and see their distinct mechanisms of action.”
We have added illustrations of the cytotoxic liposomes and gene therapy approaches.
4.3. The kappa is in our manuscript; however the conversion to the pdf file by the publisher seems to delete this letter.
4.4. “I would not refer “to cure AIDS” but attain “HIV-1 remission”
We have re-worded this sentence to read “…an important challenge in attaining HIV-1 remission, and an eventual cure.”
Reviewer 3 Report
Here, Düzgüneş and Konopka present an interesting overview of core concepts and key results regarding HIV-1 eradication strategies to kill infected cells via gene therapy, excision of integrated proviruses or cytotoxic liposomes to kill latently reversed HIV-1 cells. The review is well written and clearly organized, and it focuses in specific strategies that are not generally covered by other reviews in this very active field nowadays. Overall this makes this work interesting for a broader audience. However, my main concern is the lack of a balanced perspective presenting the critical aspects that hamper HIV-1 cure nowadays, the challenges ahead to make these strategies work in the clinic, and most importantly, the limited effect of viral reactivation agents in vivo that precludes effective HIV-1 reactivation. All these limitations, along the need for effective immune interventions, have to be clearly stated and further discussed in each section.
In particular, the section regarding “Cytotoxic Liposomes Targeted to HIV-1-Infected Cells” is specially biased as currently presented and does not offer a realistic view of the current limitations of the liposome technology and how for instance nanoparticle approaches try to improve those limitations. While liposomes/nanoparticles offer great potential, there are also several limitations that are not presented at all (i.e. difficulties of packaging hydrophobic drugs, targeted delivery to the cells of interest, cargo release, etc). Importantly, the way some ideas are written are misleading: i. e. line 200 “Liposomes can also be injected into the spinal cord (Kim et al., 1990)” are results observed in rats, not in humans, so concluding that this enables liposomes “to reach HIV-1infected macrophages/microglia in the central nervous system (Bissel & Wiley, 2004)” is not realistic for human infection. All these sentences and the animal models in which they have been tested should be clearly indicated. Overall, I think that addressing the real challenges of this particular technology and balancing possible benefits with current limitations could greatly improve the accuracy of this otherwise really well written review.
In addition, I have the following suggestions to improve the current manuscript:
1. The only effective method to eradicate HIV-1 infected cells in vivo so far is been offered by the Berlin and now the London patient examples, in which a bone marrow transplantation was required...Maybe it would be interesting to mention that despite the impossibility of applying this method to all HIV+ patients, it provides the first conceptual proof of principle that eliminating HIV infected cells can lead to an actual HIV-1 cure/remission.
2. An illustration depicting the action of LRAs, cytotoxic liposomes, suicide gene therapy and excision of Chromosome-integrated HIV-1 DNA could help to summarize approaches discussed and see their distinct mechanisms of action.
3. Kappa letter is missing in line 57
4. In line 208, I would not refer “to cure AIDS” but attain “HIV-1 remission”
Author Response
Response to Reviewers
We greatly appreciate the time the Reviewers have taken and the very useful comments they have made. We are especially grateful for the positive and collegial tone of the reviews. We have revised the manuscript to include their suggestions and to make the recommended changes. The changes in the manuscript, including the References are marked in red type.
Reviewer 3
“However, my main concern is the lack of a balanced perspective presenting the critical aspects that hamper HIV-1 cure nowadays, the challenges ahead to make these strategies work in the clinic, and most importantly, the limited effect of viral reactivation agents in vivo that precludes effective HIV-1 reactivation.”
In the revised manuscript, we have indicated some of the problems with latency reversal approaches, such as the adverse effects on CD8+ cells and non-specific inflammatory responses.
2. “effective immune interventions, have to be clearly stated and further discussed in each section.”
Our review is not focused on immune interventions, but provides alternatives to the expectation that CD8+ T cells would kill activated, infected cells. Of course, the generation of an effective vaccine (which has unfortunately been in the works since the mid-1980s, without success) may help eradicate some infected cells and prevent the infection of new cells.
3. “All these sentences and the animal models in which they have been tested should be clearly indicated.”
We have clarified the fact that the studies we have cited were done in animal models.
4.1. “…, it provides the first conceptual proof of principle that eliminating HIV infected cells can lead to an actual HIV-1 cure/remission.”
We have briefly discussed the Berlin and London patients and provided references to these cases at the end of the Cytotoxic Liposomes section.
4.2. “An illustration depicting the action of LRAs, cytotoxic liposomes, suicide gene therapy and excision of Chromosome-integrated HIV-1 DNA could help to summarize approaches discussed and see their distinct mechanisms of action.”
We have added illustrations of the cytotoxic liposomes and gene therapy approaches.
4.3. The kappa is in our manuscript; however the conversion to the pdf file by the publisher seems to delete this letter.
4.4. “I would not refer “to cure AIDS” but attain “HIV-1 remission”
We have re-worded this sentence to read “…an important challenge in attaining HIV-1 remission, and an eventual cure.”
Round 2
Reviewer 2 Report
The additional references, and caveats of the approaches added to the paper has made significant improvements and provides a more accurate picture of both the promise and challenges of current challenges to HIV cure. The new figures and diagrams are helpful in understanding the proposed approaches.
The addition of further clarification of Latency reversal drugs and their limitations is very helpful.
The major concern with Suicide gene therapy approaches is the specificity to target only infected cells, with residual non-discriminatory toxicity to other cells. Since transfection inserts the suicide gene into both infected and uninfected cells, activation will need to be very tightly controlled in cells containing the HIV genome, and not allow tat or other mechanisms to activate the gene in non-infected cells. Also not all infected cells will be transfected, leaving residual HIV- containing reservoirs as source of subsequent infection.
Line 208 “indels… can also produce replication-competent virions..” Is that not a function of the how the (CRISPR)-Cas9 target sequence is designed, or are the authors saying that this approach can inadvertently also create escape mutants that are infectious? Please clarify
I suggest toning down some of the statements in Concluding Remarks
Line 305 – change foolproof to effective
Line 306-308 – “… prevailing dogma .. that it was impossible to cure HIV/AIDS…” It is not that HIV cure is impossible, but rather that all attempts to date have been unsuccessful, with serious challenges that must be overcome first. New approaches and tools will be necessary to allow for better targeted and more effective access to the HIV reservoir, which has thus far eluded definitive identification and access for intervention.
Line 314-315 –“… Granting agencies are obviously too conservative…” This is a highly subjective statement and not appropriate for a scientific journal. Granting agencies strive to identify and fund the most promising approaches, but in absence of appropriate data and controls to demonstrate likelihood of success, often target their funds in directions most likely to provide meaningful progress.
Author Response
We thank the reviewer for his/her insightful and helpful comments on the revised manuscript.
"Line 208 “indels… can also produce replication-competent virions..” Is that not a function of the how the (CRISPR)-Cas9 target sequence is designed, or are the authors saying that this approach can inadvertently also create escape mutants that are infectious? Please clarify"
Here we cited the study of Wang et al. (2018) who observed that while HIV expression can be inhibited by the CRISPR/Cas9 system, mutant virus can also be generated. It appears that with the Non-Homologous End Joining repair system, the indel composition cannot be controlled. By contrast, with the Homology-Directed Repair system, specific introduced sequences can be integrated into the Double Strand Break sites. Thus, unless the HDR system is utilized, the indels will be random, and may generate either inactive virus or mutated virus.
"I suggest toning down some of the statements in Concluding Remarks"
We have changed "figured out" to "evolved," and we have deleted a sentence and a phrase regarding granting agencies. Both of these changes are in green type.
"Line 305 – change foolproof to effective"
We have made the requested change.
"Line 306-308 – “… prevailing dogma .. that it was impossible to cure HIV/AIDS…” It is not that HIV cure is impossible, but rather that all attempts to date have been unsuccessful, with serious challenges that must be overcome first. New approaches and tools will be necessary to allow for better targeted and more effective access to the HIV reservoir, which has thus far eluded definitive identification and access for intervention."
We are merely referring to the sentiment that we perceived in the HIV/AIDS community in the 1980s and 1990s.
"Line 314-315 –“… Granting agencies are obviously too conservative…” This is a highly subjective statement and not appropriate for a scientific journal. Granting agencies strive to identify and fund the most promising approaches, but in absence of appropriate data and controls to demonstrate likelihood of success, often target their funds in directions most likely to provide meaningful progress."
As also indicated above, we have deleted our speculation and hopes about granting agencies in the revised manuscript.